

# Role of the Quasi-Biennial Oscillation on Alleviating Biases in the Semi-Annual Oscillation

Aleena Moolakkunnel Jaison[1], Lesley J. Gray[1,2], Scott M. Osprey[1,2], Jeff R. Knight[3], Martin B. Andrews[3]

[1]Atmospheric, Oceanic and Planetary Physics, University of Oxford, Oxford, United Kingdom
[2]National Centre for Atmospheric Science, Oxford, United Kingdom
[3]Met Office Hadley Centre, Exeter, United Kingdom

*Correspondence to*: Aleena Moolakkunnel Jaison (aleena.moolakkunneljaison@physics.ox.ac.uk)

**Abstract.** Model representations of the stratospheric semi-annual oscillation (SAO) show a common easterly bias, with a
weaker westerly phase and stronger easterly phase compared to observations. Previous studies have shown that resolved and parameterized tropical waves in the upper stratosphere are both too weak. These waves propagate vertically through the underlying region dominated by the stratospheric quasi-biennial oscillation (QBO) before reaching the SAO altitudes. The influence of biases in the modelled QBO on the representation of the SAO is therefore explored. Correcting the QBO biases helps to reduce the SAO easterly bias through improved filtering of resolved and parameterized waves that contribute to
improving both the westerly and easterly phases of the SAO. The time averaged zonal-mean zonal winds at SAO altitudes change by up to 25% in response to the QBO bias corrections. The annual cycle in the equatorial upper stratosphere is improved as well. Most of the improvements in the SAO occur during the QBO easterly phase, coinciding with the period when the model's QBO exhibits the largest bias. Nevertheless, despite correcting for the QBO bias there remains a substantial easterly bias in the SAO, suggesting that westerly wave forcing in the upper stratosphere and lower mesosphere is
still severely under-represented.

## 1 Introduction

The stratospheric semi-annual oscillation (SAO) is characterized by oscillating zonal-mean wind and temperature fields with a periodicity of six months observed in the equatorial upper stratosphere and lower mesosphere. The SAO dominates equatorial variability between 0.3 hPa and 5 hPa, in the region above the quasi-biennial oscillation (QBO). Both oscillations
are proposed to influence surface weather through various pathways, including their influence on the winter polar vortex (Baldwin et al., 2001; Gray et al., 2020). However, the SAO time-averaged zonal-mean zonal wind in models shows a common easterly bias of several tens of m/s compared to observations (Smith et al., 2022).

The oscillating zonal-mean wind of the SAO consists of easterlies which are centered around the solstices and westerlies
centered around the equinoxes (Reed, 1966; Hirota, 1980). The magnitude of the peak easterlies varies around 30 m/s and




westerlies around 40 m/s (Smith et al., 2017). The SAO easterlies centered around Northern Hemisphere (NH) winter are stronger compared to that in NH summer, indicating that there is an annual component to the equatorial stratopause variability (Quiroz and Miller, 1967; Delisi and Dunkerton, 1988). The semi-annual nature of the zonal wind is visible within approximately ten degrees north and south of the equator. At higher latitudes annual variability dominates (Ray et al., 1998). The SAO easterly phase onset between 0.5 – 5 hPa occurs at approximately the same time, while the westerly phase starts at higher altitudes and propagates downward over time (Quiroz and Miller, 1967).

While the SAO dominates the equatorial upper stratosphere, the QBO is the major mode of variability in the lower-middle equatorial stratosphere, primarily occupying the altitudes from 100 hPa to 5hPa. The QBO has a mean periodicity of 28 months and is known to modulate the SAO (Smith et al., 2023). The primary driving mechanism for both phases of the QBO is wave forcing by large-scale planetary waves and small-scale gravity waves (Ern and Preusse, 2009; Ern et al., 2014). The vertically propagating waves are absorbed as a result of wave-spectra saturation or critical level filtering. Wave-spectra saturation occurs as the wave amplitude grows with height due to decreasing density and dissipate energy by wave breaking, radiative damping or turbulence (Fritts and Alexander, 2003). Critical level filtering occurs through wave breaking when the wave phase velocity approaches the speed of the background winds. The resulting transfer of momentum to the background flow at equatorial latitudes leads to the descent of the QBO phase (Lindzen and Holton, 1968; Holton and Lindzen, 1972; Plumb and McEwan, 1978; Baldwin et al., 2001).

The oscillating nature of the SAO has slightly different origins to the QBO. Similar wave damping/absorption has been identified as the major driver of the SAO westerly phase (Meyer, 1970) but involving waves with faster phase speeds that can propagate higher into the atmosphere. However, the easterly phase is primarily attributed to meridional advection of summer hemisphere easterly zonal winds associated with the large-scale Brewer-Dobson circulation (BDC) (Dobson et al., 1929; Brewer, 1949; Butchart et al., 2014). While the westerly wave forcing of the SAO is present throughout the year, the BDC is forced by extra-tropical wave driving that is strongest during the winter of each hemisphere. This gives rise to the semi-annual nature of the oscillation (Holton and Wehrbein, 1980). The different strengths of the Southern Hemisphere (SH) and Northern Hemisphere (NH) winter BDC (NH BDC being stronger), also gives rise to the annual cycle within the SAO region (Quiroz and Miller, 1967; Delisi and Dunkerton, 1988).

Models, reanalyses and observational data sets have been used to understand the QBO and the SAO and to examine their relationship (e.g. Burrage et al., 1996; Dunkerton and Delisi, 1997; Garcia et al., 1997; Ray et al., 1998; Garcia and Sassi, 1999; Richter and Garcia, 2006; Peña-Ortiz et al., 2010; Smith et al., 2017; Smith et al., 2023). The modulation of the westerly phase of the SAO by the QBO is widely acknowledged. Garcia et al. (1997) and Dunkerton and Delisi (1997) have shown using rocket-sonde observations that the altitude of maximum descent of the westerly SAO can be modulated by the QBO. Later studies have confirmed this result using global models (Peña-Ortiz et al., 2010) and satellite data (Ern et al.,



2015; Smith et al., 2023). Smith et al. (2023) further examined satellite observation data and found that the QBO modulates
not only the depth but also the magnitude of the SAO westerly phase, with an almost 10 m/s increase during the QBO
easterly (QBOE) phase. The generally acknowledged mechanism of this QBO influence on the SAO is wave filtering. Smith
et al., (2023) showed that the differences in the SAO winds in the upper stratosphere due to the phase of the QBO are
confined to the low latitudes which led them to suggest that the QBO influence is mainly through vertical wave coupling, in

agreement with prior studies that have shown that both resolved waves and parameterized GW reaching SAO altitudes
depend on the wind profiles at QBO altitudes (Garcia et al., 1997; Peña-Ortiz et al., 2010).

Early studies using rocket-sonde observations did not find a convincing relationship between the QBO and the easterly phase
of the stratospheric SAO (Dunkerton and Delisi, 1997; Garcia et al., 1997). Since the SAO easterly phase is widely accepted

to be driven by meridional advection associated with the BDC, an absence of direct vertical coupling from the QBO in this
phase is not surprising. However, Peña-Ortiz et al. (2010) found a modulation of both the strength and altitude of maximum
descent of the SAO easterly by the QBO in their model analysis using MAECHAM. While this is consistent with some
reanalysis studies that identified the presence of a QBO signal in the upper stratosphere during NH winter, suggesting a
QBO modulation of the SAO easterly phase (Pascoe et al., 2005; Calvo et al., 2007; Peña-Ortiz et al., 2008), the paucity of

validatory observations means that overestimation of the influence of resolved and small-scale waves by the models
(employed by both Peña-Ortiz et al. 2010 and in the generation of the reanalysis products) cannot be excluded. Later, Ern et
al. (2015) noted that in one of their case studies using reanalysis and satellite data during times when the QBO filtering of
westward waves was minimal, westward waves were found to travel through to the upper stratosphere and the SAO easterly
phase exhibited a downward propagation, suggesting a modulation of the phase descent by the vertically-propagating waves.

However, Ern et al. (2015) also questioned the reliability of this result. In their study of SABER satellite data Smith et al.
(2023) concluded that the QBO primarily affects the SAO westerly phase rather than the easterly phase. Additionally, no
evidence for a QBO modulation of the SAO easterly phase component of the annual cycle has been reported.

Another mechanism through which the QBO could influence the SAO easterlies is via the extra-tropics. The QBO is

generally believed to influence mid latitude Rossby wave propagation through the Holton-Tan mechanism (Holton and Tan,
1980; Anstey and Shepherd, 2014; Anstey et al., 2022). The BDC is strengthened during extreme events known as Sudden
Stratospheric Warmings when the polar vortex is substantially weakened or destroyed as a result of the transfer of easterly
momentum from large-scale Rossby waves to the zonal flow at mid- and high-latitudes (Baldwin et al., 2019). This
momentum transfer in turn strengthens the BDC and hence the cross-equatorial flow that generates the SAO easterlies

through meridional advection. The frequency and timing of SSWs is known to be sensitive to the QBO (Gray et al., 2004;
Pascoe et al., 2006; Anstey et al., 2022) and thus will likely influence the strength and timing of the incoming easterly SAO.
There is also evidence that the QBO influences the height of the maximum cross-equatorial flow (Lu et al., 2020, see their
Fig. 11) which may therefore influence the depth to which the SAO easterlies penetrate.  However, none of the previous



studies have examined the impact of this mechanism in detail since, as mentioned above, the QBO modulation of the easterly
SAO phase appears to be much smaller than its modulation of the westerly SAO phase.

The Quasi-Biennial Oscillation initiative (QBOi) project (Anstey et al., 2020) aims to examine and improve the representation of QBO in models. Butchart et al. (2020) analyzed the simulated QBO in various global climate models and highlighted that most models have an easterly phase QBO (QBOE) that is generally too weak and exhibits a westerly time
mean wind bias throughout the depth of the QBO (see also Rao et al., 2020; Garfinkel et al., 2022). Through the various mechanisms described above, this bias in the underlying QBO could influence the upper stratosphere and thus lead to a bias in representation of the SAO. In this study we employ a climate model to explore this possibility. The modelled QBO wind biases are corrected by nudging the zonal-mean zonal wind in the equatorial low-mid stratosphere towards reanalysis data. The representation of the SAO is then examined to determine whether biases in the SAO have been improved. The paper is
structured as follows: Section 2 outlines the techniques employed in the study, including a description of the model and datasets. Results are presented in Section 3, and Section 4 summarizes the findings.

## 2 Data and Methodology

### 2.1 Model and Experimental set up

The model data used in this study are from simulations of the HadGEM3 GA7.1 N216 atmosphere-only model, performed as part of the UK contribution to phase 2 of the QBOi project. The QBOi project aims to improve the understanding and representation of tropical stratospheric variabilities in climate models (see Butchart et al. 2018 for a description of the over-arching aims of the project). The model is based on the Met Office Hadley Centre AMIP model used for the Coupled Model Intercomparison Project Phase 6 (CMIP6) historical runs (Eyring et al., 2016). The N216 horizontal resolution has 0.54 x
0.83 degrees latitude-longitude horizontal resolution (approx. 60 km) and 85 vertical levels extending to 85 km (0.01hPa). Observed sea-surface temperature and sea ice distributions from the CMIP6 specification were imposed at the lower boundary. The CMIP6 historical forcings were used until 2014 and CMIP6 SSP5-85 forcings were used from 2015-2020. The only difference from the CMIP6 set-up was the use of climatological ozone instead of the time-varying values. The gravity wave scheme has also been updated to include convective coupling in the non-orographic parametrization (Bushell et
al., 2015).

The simulations extend from January 1979 to December 2020 (42 years). Two experiments were analyzed, the 'Control' and 'Nudged' experiments. Three ensemble members were performed for each experiment. In the Control experiment the model was allowed to evolve freely and in the Nudged experiment the equatorial stratospheric zonal-mean zonal winds were
nudged towards the European Centre for Medium-Range Weather Forecasts (ECMWF) ERA5 zonal-mean field in the height





region of the QBO. The nudging methodology followed the Stratospheric Nudging And Predictable Surface Impacts (SNAPSI) protocol (Hitchcock et al., 2022), although the nudged regions and timescale differ. The nudging was fully applied between 10-70 hPa with gradual tapering to zero by 100 hPa and 5hPa, and between 10°S-10°N, with tapering to zero by 20° latitude. This ensured that the nudging was applied only to the QBO region and the zonal winds in the SAO

region were allowed to evolve freely. The nudging time scale used was 5 days, which is expected to be sufficient to constrain the slowly evolving QBO winds. To mimic the effect of momentum transfer from wave damping/absorption, only the zonal winds were nudged, so the temperatures and meridional winds were able to respond to the zonal wind distributions. Additionally, only the zonal-mean of the zonal winds was nudged, to allow waves to evolve freely and thus avoid any significant artifacts (Hitchcock and Haynes, 2014; Hitchcock and Simpson, 2014; Martin et al., 2021; Hitchcock et al.,

2022). The nudging is introduced into the model as an additional forcing term in the zonal momentum equation of the form - $\alpha$ ($\bar{u}$ - $u_{ana}$) where $\bar{u}$ is the zonal mean zonal wind, $u_{ana}$ is the target state and $\alpha$ is the relaxation parameter equal to the inverse of the 5-day nudging time scale. In all other respects the Nudged experimental set-up was identical to the Control experiment.

The HadGEM3 model includes a spectral gravity wave parametrisation that represents the effects of non-orographic gravity waves with horizontal and vertical scales smaller than the model resolution (Warner and McIntyre, 1996, 1999, 2001; Scaife et al., 2000, 2002). An isotropic spectrum of gravity waves is initiated close to the earth's surface at ~ 400 m. The waves propagate vertically until they are dissipated by critical level filtering and saturation. The amplitude of the gravity wave source is proportional to the square root of total precipitation to capture the spatial and temporal variability of their

generation.

## 2.2 Reanalyses

In this study, two reanalysis datasets were utilized. Firstly, the European Centre for Medium-Range Weather Forecasts (ECMWF) ERA5 reanalyses (Hersbach et al., 2020) was used in the nudging scheme to ensure a good representation of the QBO, in accordance with the QBOi protocol. ERA5 is produced using the ECMWF Integrated Forecast System (IFS) 41r2.

The dataset from 1979 is used to match the length of AMIP runs. ERA5 has a horizontal and vertical resolution of T639 (~31km) and 137 hybrid sigma-pressure model levels respectively, extending to 0.01hPa. The data are interpolated on to a lower resolution grid for nudging purposes (for further details of the relaxation methods, see Knight et al., 2021).

Although ERA5 has a good representation of the QBO compared to observations (Ern et al., 2023) the SAO winds in ERA5

are unrealistic, with westerly phase magnitudes reaching as high as 150m/s (Shepherd et al., 2018; Ern et al., 2021) which are much larger than estimates of 40 m/s from satellite derived winds (Smith et al., 2017). For this reason, the Modern-Era Retrospective Analysis for Research and Applications, Version 2 (MERRA-2) dataset was employed for comparison with the model results in the SAO region. The MERRA-2 reanalysis dataset (Gelaro et al., 2017) is a global reanalysis dataset



provided by the National Aeronautics and Space Administration (NASA) Goddard Institute for Space Studies (GISS). The
dataset is available from 1980 and has a horizontal resolution of 0.5° latitude × 0.625° longitude and a vertical resolution of
72 hybrid-eta model levels extending from the surface to 0.01 hPa. Previous research has identified MERRA-2 as the
reanalysis dataset most similar to observations in the SAO region (Ern et al., 2021). This may be due to the assimilation of
Microwave Limb Sounder (MLS) temperature data at altitudes of 5hPa and above, along with a non-orographic gravity wave
parameterization tuned to better represent equatorial stratospheric variability (Molod et al., 2015).

**2.3 TEM Diagnostics**


The Transformed Eulerian mean momentum equation (Andrews et al., 1987) was used to analyze the processes driving the
SAO.

$$\frac{\partial \overline{u}}{\partial t} = -\overline{v}^* \left[\frac{1}{a\cos\phi}\frac{\partial(\overline{u}\cos\phi)}{\partial\phi} - f\right] - \overline{w}^*\frac{\partial\overline{u}}{\partial z} + \frac{1}{\rho_0 a\cos\phi}\nabla\cdot F + \overline{X},\tag{1}$$

where


$$\overline{v}^* = \overline{v} - \frac{1}{\rho_0}\left(\frac{\rho_0\,\overline{v'\theta'}}{\overline{\theta}_z}\right)_z$$

$$\overline{w}^* = \overline{w} + \frac{1}{a\cos\phi}\left(\frac{\cos\phi\,\overline{v'\theta'}}{\overline{\theta}_z}\right)_\phi$$

$$\nabla\cdot F = \frac{1}{a\cos\phi}\frac{\partial}{\partial\phi}\left(F^{(\phi)}\cos\phi\right) + \frac{\partial F^{(z)}}{\partial z}$$

and

$$F^{(\phi)} = \rho_0 a\cos\phi\left(\overline{u}_z\frac{\overline{v'\theta'}}{\overline{\theta}_z} - \overline{v'u'}\right)$$


$$F^{(z)} = \rho_0 a\cos\phi\left(\left[f - \frac{1}{a\cos\phi}\frac{\partial(\overline{u}\cos\phi)}{\partial\phi}\right]\frac{\overline{v'\theta'}}{\overline{\theta}_z} - \overline{w'u'}\right)$$

Here $u$, $v$, $w$ and $\theta$ represent the zonal wind, meridional wind, vertical wind and potential temperature. The overbar denotes
a zonal mean quantity, and the prime denotes departure from the zonal mean. The model output provides pre-calculated
TEM variables. TEM diagnostics in reanalysis is calculated using the three-hourly averaged $u$, $v$, $w$ and temperature fields.
The calculations are performed in pressure coordinates and transformed into log-pressure coordinates to obtain the
formulation of Andrews et al. (1987) shown above (see Gerber and Manzini, 2016 and their corrigendum). The acceleration
of zonal-mean zonal wind is determined by four distinct forcing components in the TEM formulation. The first term on the
right-hand side of equation (1) indicates meridional advection. In the context of the SAO this primarily consists of easterly



winds that are transported by the BDC from the summer to the winter hemisphere at SAO altitudes. The second term represents vertical advection i.e. vertical advection by the BDC or due to a local wave-driven circulation. The next two terms contribute to the wave-driving forces: the third term is the Eliassen-Palm (E-P) flux divergence, which indicates the resolved wave forcing and the fourth term X represents the remaining sub-grid scale processes. In the model, X mainly consists of the parameterized gravity wave drag and numerical diffusion.

All diagnostics presented in this paper were compiled using individual ensemble-members before an average of the three ensemble members was taken. Unless otherwise stated, the average was found to be similar to results from the individual members. Wherever a comparison is made between the Control and Nudged experiment results, the differences are assessed for statistical significance using a two- sided Student's t-test. The null hypothesis states that the Control and Nudged data are drawn from the same statistical distribution and have identical (population) averages. A p-value less than 5% is considered statistically significant, i.e., there is a statistically significant difference between the Control and Nudged means.

## 3 Results

### 3.1 SAO bias alleviation

The climatology of zonal-mean zonal wind from MERRA-2 is displayed as a time-height cross section in Fig. 1a. As expected, it shows alternating westerlies and easterlies forming the SAO spanning from 5 hPa to 0.1 hPa with an approximate period of 6 months. The SAO in MERRA-2 at 1 hPa aligns closely with the satellite-derived wind magnitudes from the SABER and MLS satellite data shown in Smith et al. (2017) Fig. 4, particularly during the easterly phase, although there are some small differences in the westerly phase amplitudes, with SABER and MLS indicating westerly SAO magnitudes around 20-25 m/s, whereas MERRA-2 shows magnitudes reaching up to 30 m/s.

The corresponding time-series of the HadGEM3 control simulation is displayed in Fig. 1b. The model exhibits a pattern that is consistent with the observed SAO. However, a clear distinction in the amplitude and duration of each phase between Fig. 1a and b is evident, with an overall easterly bias evident in the model, consistent with that found by Smith et al. (2022). The easterly SAO phase, particularly during JJA near 1hPa, is much stronger in the HadGEM3 Control with a strength difference of 30 m/s compared to the reanalysis. The MERRA-2 easterly phase is clearly weaker in JJA (SH winter) than in DJF (NH winter) but this imbalance is much weaker in the Control experiment, leading to a weaker annual cycle. Both phases of the Control westerly SAO phase also show an easterly bias of at least 25 m/s at 1hPa.

To assess changes in the SAO characteristics between the Control and Nudged experiments, the corresponding climatology from the Nudged experiment is shown in Fig. 1c. It shows a promising reduction of the easterly bias of the modelled SAO.





The magnitude of the SAO westerly phase has increased in both equinoxes. Most notably, the easterly phase in JJA (SH winter) appears slightly reduced in magnitude around 1hPa and the westerly phase in MAM is strengthened, which will improve the annual cycle as well.

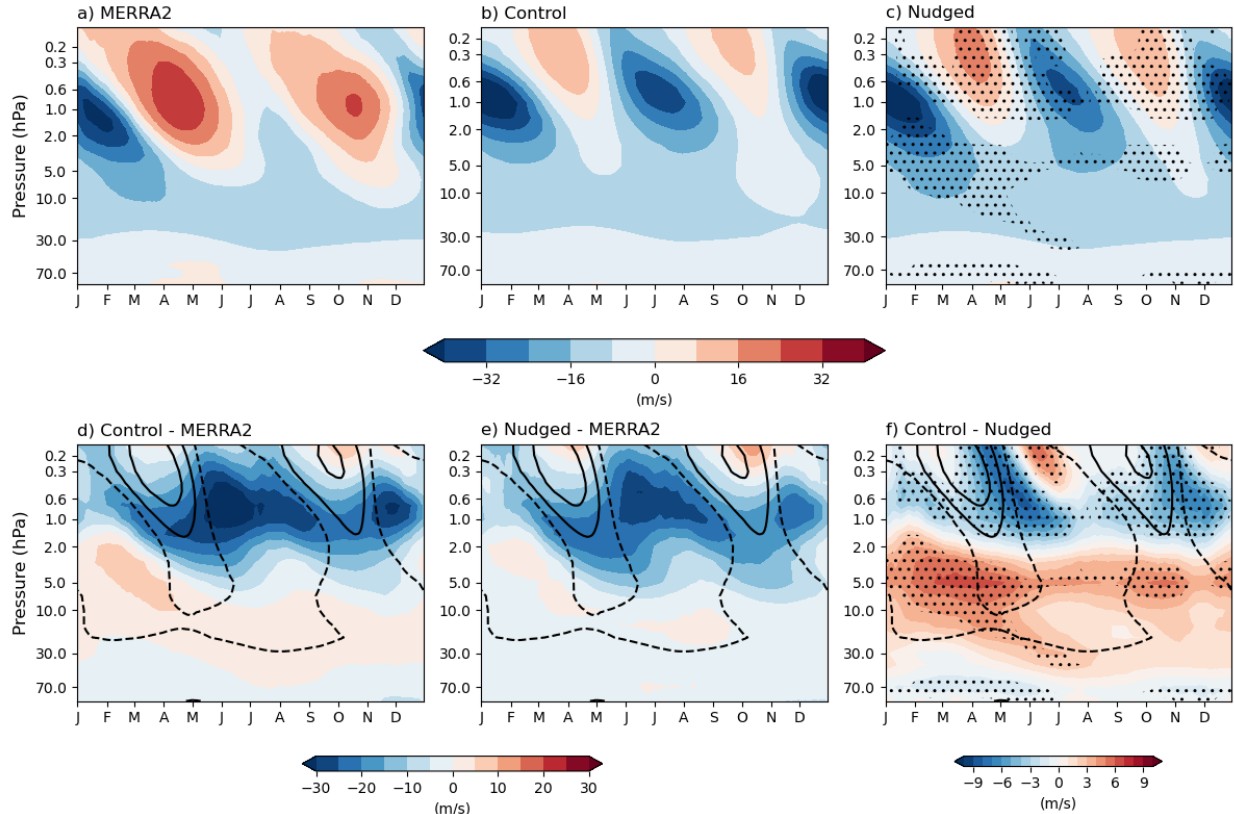

**Figure 1: Daily mean climatology of zonal-mean zonal wind (ms⁻¹) averaged over 15°N to 15°S for (a) MERRA-2, (b) Control ensemble-mean and (c) Nudged ensemble-mean, (d) Control minus MERRA-2 differences, (e) Nudged minus MERRA-2 differences and (f) Control minus Nudged differences. Overlaid are the wind contours from the Control ensemble-mean at 10, 0 and 10m/s. Stippling denotes 95% confidence interval. Stippling in (c) is the same as in (f) to easily identify where the nudged experiment is statistically different from control.**


To confirm this improvement the Control-minus-MERRA-2 and the Nudged-minus-MERRA-2 differences are shown in Fig. 1d-e, noting that negative (positive) values indicate stronger easterlies (westerlies) than MERRA-2. Comparison of the figures confirms that the westerly bias in the QBO region from around 70 hPa to 5 hPa in the Control experiment has been eliminated by the nudging, as expected. (It also confirms that nudging towards ERA5 data in this region while using 235 MERRA-2 as validation data is acceptable, since the two datasets are almost identical in this height region). Above that





region, although there is still clearly an easterly bias in the SAO in the Nudged experiment, the amplitude of the bias has been reduced.

The nature of the bias reduction can be seen more clearly in Fig. 1f which shows the Control-minus-Nudged differences. The
reduction in easterly bias occurs almost throughout the year between 1 hPa and 0.3 hPa, reaching its maximum difference of about 8 m/s during the SAO phase transition from westerly to easterly (Apr-Jun, Oct-Dec). The timing of this maximum bias reduction can be attributed to a slightly prolonged SAO westerly phase duration and a delayed onset of the easterly phase in the Nudged experiment. However above 0.3hPa, the easterly phase magnitudes increase faster in the Nudged compared to the Control experiment and this is seen as positive values at these altitudes in June and December in the difference plot (Fig.
1f).

In summary, Fig. 1 demonstrates that decreasing the wind biases in the lower stratosphere improves the SAO representation, although the biases are not completely alleviated. A time mean of zonal mean zonal wind from 15°S to 15°N for all 42 years (not shown) indicates that the maximum SAO wind correction between 2 hPa to 0.6 hPa reaches up to 25%. At each altitude,
the ratio of difference in Control and Nudged winds to the Control winds is used to calculate the percentage of wind change at that altitude. While the winds from the Nudged experiment align more closely with the reanalysis there remains considerable room for improvement. In particular, the easterly phase during JJA is still too strong, while the westerly phase is too weak and does not extend far enough downward. Nevertheless, alleviation of the QBO bias has clearly improved the simulation of the SAO. In the remaining sections of the paper, we therefore examine in more detail how the QBO corrections
have led to a reduction in SAO bias.

## 3.2 QBO modulation of SAO and biases

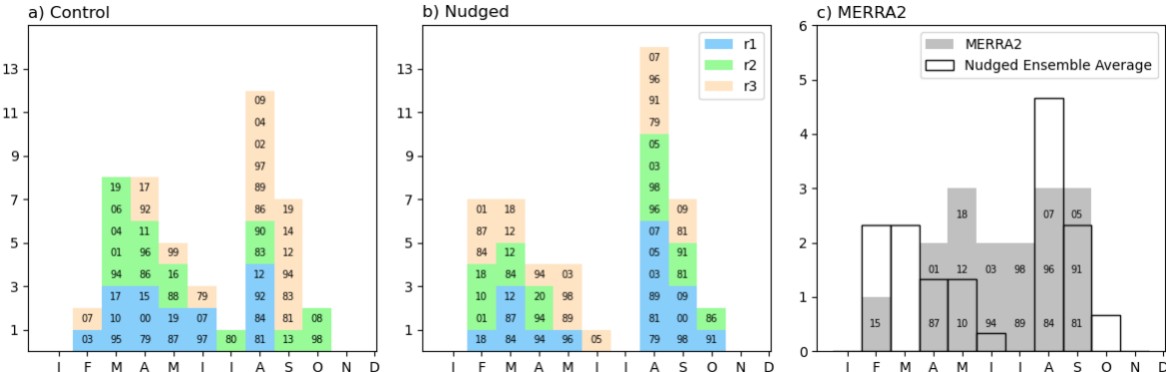

**Figure 2: Histogram of QBOE to QBOW phase transition months at 5hPa for the (a) Control and (b) Nudged ensemble members
and (c) at 10 hPa for MERRA-2. Corresponding years for the (a) Control and (b) Nudged and (c) MERRA-2 are listed. In figures
a-b the colours denote the different ensemble-members. Averages of the three Nudged run ensemble members are overlaid in (c).**



Examining a QBO composite can help to gain a better understanding of how the SAO is impacted by correcting the QBO winds and how the QBO influence extends to the upper stratosphere. The QBO composite is calculated using the following method. Firstly, the QBOE-to-QBOW transition months are found using the raw monthly-mean zonal-mean zonal wind data averaged over 15S to 15N. The months in which the QBO transitions from easterlies to westerlies at 5hPa are identified. Moreover, the zonal wind average for the next 4 months is required to be westerly to avoid counting occasional SAO westerly phases descending to 5hPa, without concomitant QBO westerlies. Fig. 2a and b show the QBO transition months which satisfy these conditions for the Control and Nudged simulations respectively (using different colours for the three ensemble-members). Depending on whether the month is closer to the NH spring equinox (i.e. between Jan-Jun) or autumn equinox (between Jul-Dec), data for 1000 days starting from either March 1st or September 1st is extracted and used to form the composite. A 1000-day time-series has been chosen so that the full QBO cycle can be captured, since the QBO period can typically range up to 34 months (Baldwin et al., 2001). It is important to note that the Fig. shows an average over many QBO cycles. Especially in the QBO region, the cycle-to-cycle variability in duration and depth of each QBO phase, along with the generally higher magnitude of the QBOE phase compared to the QBOW phase affects the compositing. Nevertheless, using this composite allows the visualization of, for example, the evolution of the SAO as the westerly phase of the QBO descends.

In Fig. 1, it was noted that the overall westerly bias in the QBO region is eliminated through nudging. Figure 3 illustrates the impact of nudging on each QBO phase. In the QBOW phase the QBO westerlies of the Control experiment (Fig. 3a) are stronger and last longer at altitudes around 5 hPa and as the westerlies propagate downward they become weaker than in the Nudged experiment (compare above and below 30 hPa). In the Nudged experiment (Fig. 3b), the QBOW phase strength and duration is roughly the same throughout the QBO altitude range. This suggests possibly excessive eastward wave momentum deposition at higher altitudes in the Control experiment. The QBOE phase on the other hand appears to be weaker throughout the altitude range 70-10hPa in the Control experiment compared to the Nudged experiment, confirming the westerly bias of the QBO in the free-running model.

Focusing on altitudes above the 2 hPa level where the maximum in SAO amplitude is found (see Fig. 1), both the Control and Nudged experiments clearly indicate that the strength of the SAO is modulated by the QBO (Fig. 3a, b). As the QBOW phase progresses downward, the depth to which the SAO westerly phase extends also increases, finally merging with the next QBOW phase. In contrast to the observations (Garcia et al., 1997; Smith et al., 2023), the SAO easterly phase also appears to be modulated by the QBO in the simulations. The SAO easterly phase is stronger and lasts longer when QBO westerlies are present around 10 hPa (i.e. when the QBO at 50 hPa is in its easterly phase).



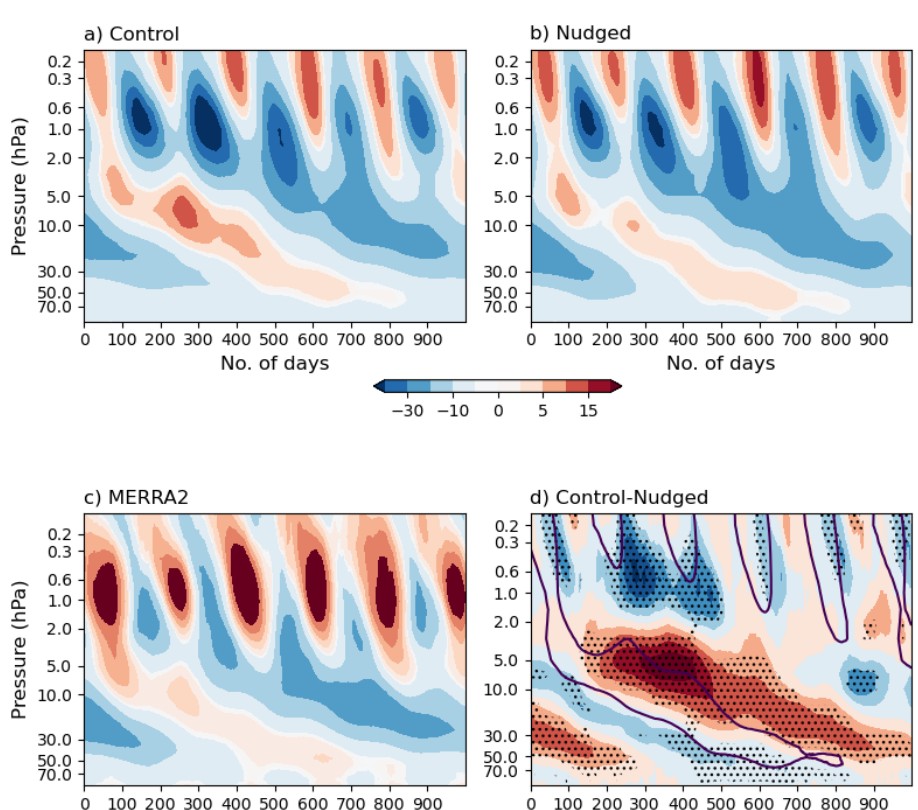

**Figure 3: QBO composite of zonal-mean zonal wind (m/s) for the (a) Control experiment, (b) Nudged experiment and (c) MERRA-2. The composite members start from 1st of March or 1st of September whichever is closest to the start of the QBOW phase. The 5hPa reference level is used to identify the start of the QBO westerly phase in model and 10 hPa is used in MERRA2 (see text for more details). (d) Difference between the Control and Nudged experiments, with the QBO composite winds from the Control experiment overlaid as black contours. Stippling denotes 95% confidence interval. Stippling denotes 95% confidence interval.**

Figure 3c shows the corresponding MERRA-2 QBO composite of zonal-mean zonal winds except that the QBO transition month has been selected based on the sign of the winds at 10 hPa instead of 5 hPa since the SAO westerlies in MERRA2 frequently extend to levels below 5 hPa. Figure 2c shows the QBOW to QBOE transition months chosen for the composite in MERRA-2. We note that there is only one MERRA-2 ensemble member, so the MERRA-2 QBO composite has been calculated using only 16 1000-day samples whereas the model experiments have 3 ensemble-members and each of the experiment composites contains 45 1000-day samples. Although the reanalysis has stronger westerlies and weaker easterlies compared to control in the SAO region, as already seen in Fig. 1, the characteristics of the SAO modulation by the QBO are nevertheless quite similar in those displayed by the model.




Figure 3d shows the Control-minus-Nudged differences between the QBO composites and illustrates in more detail how correcting the QBO biases has affected the SAO. Consistent with previous findings (Fig. 1d), the differences maximise during the transition from QBO westerlies to QBO easterlies and in the months when the SAO transitions from westerlies to easterlies. The figure highlights how complementary the improvements in the QBO and the SAO are to each other.

Irrespective of the SAO phase, whenever a correction is made to the westerly bias at QBO altitudes, a corresponding easterly bias correction occurs at the SAO altitudes. For instance, between days 200 and 400, the most statistically significant changes in the QBO occur at 5hPa, coinciding with the most pronounced easterly bias reduction in the altitudes of the SAO. During this period, the QBO westerlies diminish while the easterlies intensify. It is noteworthy that the SAO easterlies and QBO easterlies converge at 5hPa. Meanwhile, the SAO westerly (easterly) phase around days 200 to 400 at 2hPa and higher

are stronger (weaker) in the Nudged experiment (Fig. 3b) compared to the Control experiment (Fig. 3a) and extends further down in altitude. It is noted that the most significant SAO corrections occur when the lower stratospheric QBO winds at 50 hPa are easterly and the mid stratospheric QBO winds at 10hPa are westerly (both roughly coincide).

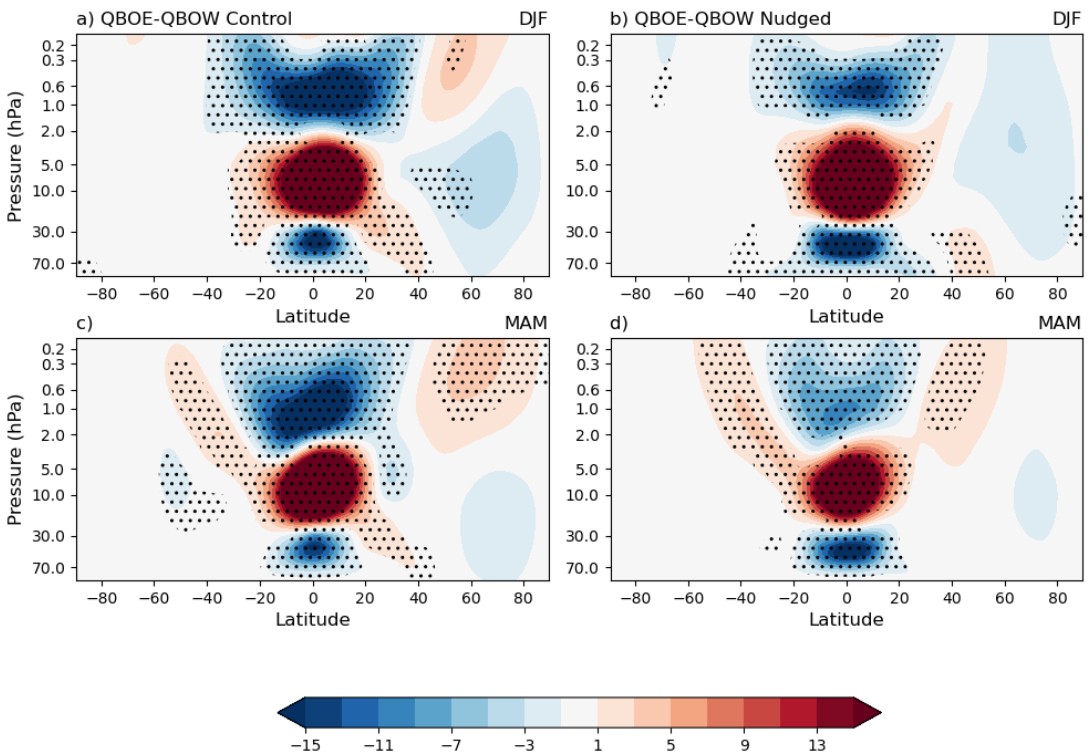

**Figure 4: Latitude-height seasonal composites of QBOE minus QBOW zonal-mean zonal winds (m/s) from a) Control experiment DJF, b) Nudged experiment DJF, c) Control experiment MAM and d) Nudged experiment MAM. The phase of the QBO has been defined at 50 hPa. Stippling denotes at 95% confidence interval.**





Figure 4 shows the latitude-height cross section of QBOE-minus-QBOW composite differences in zonal-mean zonal winds
for both the Control and Nudged experiments during DJF and MAM (results are similar for JJA and SON as well). In
constructing these composite differences, the QBO phase is determined by whether the 50 hPa QBO winds are westerly (>2
m/s) or easterly (<-2 m/s). The additional benefit of choosing 50 hPa QBO winds as a reference is that the observed Holton-
Tan (HT) relationship is associated with the winds at this level, and any changes associated with the HTE might also be
evident. Figure 4 reaffirms that during the QBOE phase at 50 hPa the SAO easterlies in DJF are stronger than during the
QBOW phase (negative values in Fig. 4a,b from 2 hPa to 0.2 hPa at the equator) and the SAO westerlies in MAM are
weaker (negative values in Fig. 4c,d from 2 hPa to 0.2 hPa at the equator). When the 50 hPa QBO winds are easterly most of
the stratosphere i.e., from 30 hPa to 5 hPa is occupied by westerly winds, thus filtering out more eastward travelling waves
and allowing more westward travelling waves to pass through. Another notable result is that the SAO wind strength
difference between QBOE and QBOW is clearly reduced in the Nudged experiment compared to the Control experiment.
This reduction can be attributed to SAO bias corrections (reducing the SAO easterly bias) primarily occurring during QBOE,
hence reducing the difference in SAO during QBOE and QBOW.

Both the Control and Nudged experiments show a weak H-T relationship, with weaker DJF westerlies at mid-to-high
latitudes during the QBOE phase (see Elsbury et al., 2021). The sign of the differences near the equatorial stratopause SAO
region is consistent with this, since weaker mid-latitude winds suggest stronger planetary wave forcing of the BDC.
However, determining whether the changes to the SAO have arisen as a result of the filtering of vertically propagating waves
at the equator or as a result of changes to the BDC is not possible from these zonal wind diagnostics, and requires a more
detailed examination of the various contributions to the momentum equation (see next section).

**3.3 Forcing terms**

The climatological height-time evolutions of the four forcing terms in equation (1) are shown in Fig. 5. The average behavior
of all four terms is similar in both the experiments. Above 1 hPa both the experiments indicate that (i) meridional advection
is the major westward forcing term (Fig. 5a,b), (ii) GWD and vertical advection provide, on average, eastward forcing (Fig.
5d,e and 5j,k), and (iii) eastward forcing by EPD is strongest at the equinoxes (Fig. 5g,h). This is consistent with earlier
research which found that the westerly SAO phase is primarily driven by wave forcing and the easterly phase is largely
driven by meridional advection (Meyer, 1970; Holton and Wehrbein, 1980).

The Control-minus-Nudged differences in the final column in Fig. 5 outline the changes in the four forcing terms as a result
of the bias corrections in the lower stratosphere zonal-mean zonal winds. We note that negative (positive) values indicate





more eastward (westward) forcing in the Nudged experiment, and that increased eastward forcing is desired to reduce the easterly bias of SAO.

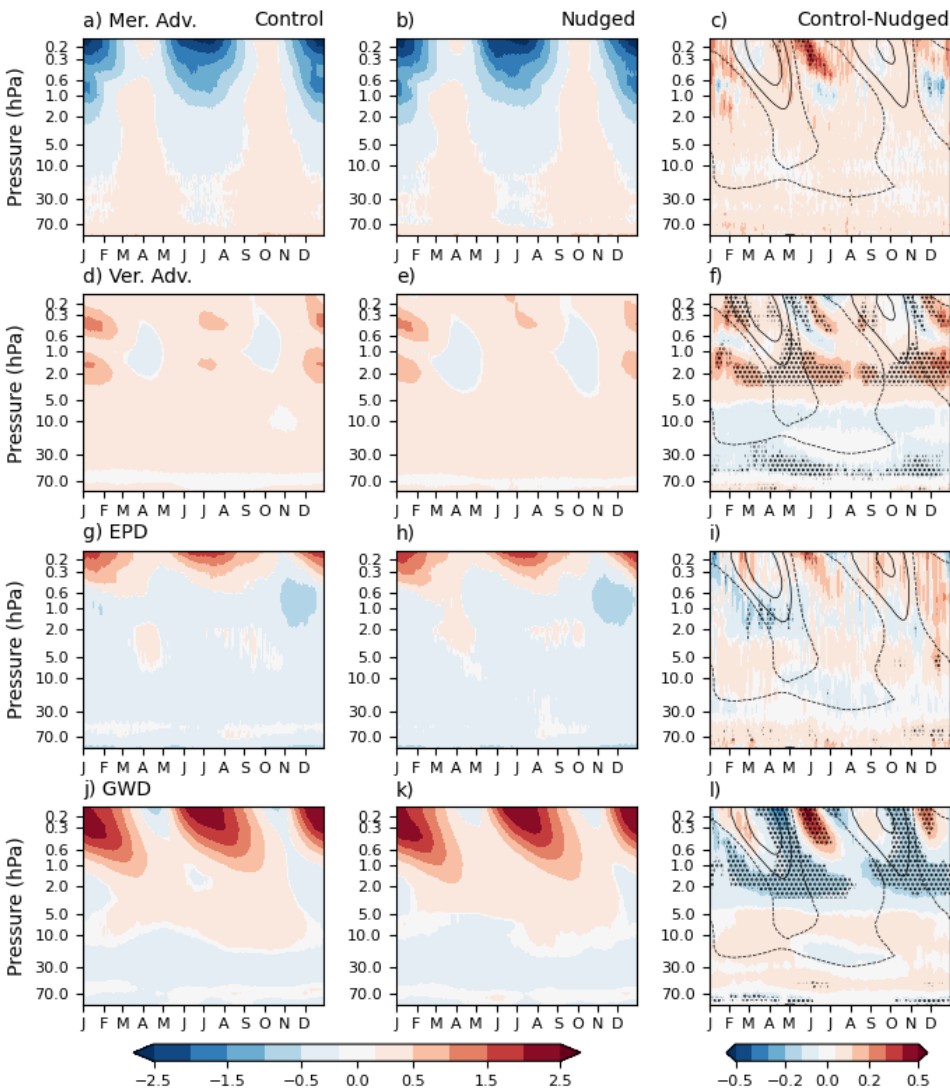

**Figure 5. Climatology of TEM forcing terms averaged over 15S-15N. (a) and (b) show the forcing due to meridional advection in**
**the Control and Nudged ensembles respectively. (c) shows the corresponding Control-minus-Nudged differences. Similarly (d-f)**
**shows the corresponding forcing due to vertical advection, (g-i) EPD and (j-l) GWD. Units are m/s/day. Zonal-mean zonal wind**
**contours of -10, 0, 10 m/s are overlaid on (c), (f), (i) and (l). Solid contours denote westerlies and dashed counters denote easterlies.**
**Stippling denotes 95% confidence interval.**



The strength of the meridional advection forcing in Nudged shows a shift in altitude compared to Control, which is evidenced by the dipole structure in DJF and JJA above 2hPa in Fig. 5c. However, the difference is statistically significant only in JJA above 0.6 hPa where the increase in westward meridional advection occurs. This increased westward forcing may explain the easterly phase magnitudes increasing faster in the Nudged compared to the Control experiment, above 0.6 hPa, as noted in the discussion of Fig. 1.


Vertical advection does little to reduce the SAO easterly bias. The vertical advection forcing is mostly present when there is a strong vertical gradient in zonal wind (compare Fig. 5 d,e to Fig. 1). Between 1.0-0.6 hPa the SAO winds reach their maxima so they have the least vertical gradient and thus vertical advection forcing at these levels is minimal. The difference plot in Fig. 5f, between 1.0-0.6 hPa confirms that changes associated with vertical advection changes have minimal impact at

these altitudes. This coincides with the levels where the reduction in the SAO easterly bias is maximum (Fig. 1). Thus, the opposing force from vertical advection has minimal impact between 1.0-0.6 hPa. The maximum change in vertical advection is seen at altitudes between 1 hPa and 2 hPa. This is consistent with the changes in wave-driven induced circulation as the GWD shows an increased eastward forcing at these altitudes. However, these vertical advection changes are outside the altitude range of the maximum SAO bias correction.


There is some evidence for a statistically significant increase in eastward wave forcing associated with resolved waves during the SAO transition from its easterly to westerly phase (Fig. 6g-i), which compares better with corresponding diagnostics from the MERRA2 reanalysis (see Jaison et al., 2024 (in production), Fig. 7). Starting at 0.3hPa and descending to 2hPa from January to May, the improvement in EPD appears as negative values close to the zero-wind contour in Fig. 5i,

aiding a faster transition to the SAO westerly phase. Similar, but weaker, improvements can be seen in Aug-Nov as well. In the QBO region the waves are damped along the strongest shear zones (Pahlavan et al., 2021), thus aiding the phase transition. The improvement in resolved wave forcing here implies more wave damping along the strongest westerly shear zones, thus aiding the faster phase transition to SAO westerlies. Additionally, during the SAO westerly phase months, especially during MAM below 0.6 hPa (Fig. 5i), there is an increase in eastward wave forcing, which will act to strengthen

the SAO westerly phase.

The variations in GWD are shown in Fig. 5 j–l. One notable feature is that according to Fig. 5l, the most significant changes in GWD occur during the transition from the westerly to easterly SAO phase (Apr-May, Oct-Nov), precisely during the months when the zonal-mean zonal wind improvement is at its highest (Fig. 1f). A westward GWD forcing is visible at SAO

altitudes in the Control experiment during this time, as shown in Fig. 5j, and the forcing is noticeably reduced in the Nudged experiment where eastward forcing prevails (Fig. 5k), helping to increase the SAO westerly phase duration in the Nudged



experiment. At altitudes below 1 hPa, the improvements associated with the GWD forcing are compensated by the increased eastward forcing in vertical advection, while above 1hPa vertical advection has little to no effect on the GWD improvements.

In summary, the analysis indicates that in the Nudged experiment both meridional advection and GWD contribute to diminishing the magnitude of the easterly phase of the SAO at around 1 hPa, whilst the wave forcing terms EPD and GWD act to improve (increase) the amplitude and duration of the westerly phase.

**3.4 QBO modulation of TEM variables**

Figure 6 explores the influence of the individual QBO phases on modulating the time-mean SAO forcing terms and how the forcings are affected by the nudging of the QBO. The forcing terms are calculated for the QBOE and QBOW phase chosen depending on whether the zonal-mean zonal wind at 50 hPa is less than –2 m/s or greater than 2 m/s respectively (the QBO index is determined for each month independently and then the diagnostics are composited into a time-mean; it is therefore not an annual-mean). The evolution of the composite zonal winds during both the QBO phases are shown in Fig. 6a,b. When

the winds are easterly at 50 hPa they reverse to westerly at around 10 hPa. When the winds are westerly at 50 hPa, they reverse to easterlies at around 30 - 10 hPa. Above the 10 hPa level, where the SAO dominates, the averaged winds are easterly in both QBO phases which is a manifestation of the easterly bias in the model SAO.

For comparison, MERRA-2 QBOE and QBOW composites of zonal-mean zonal wind and TEM diagnostics are shown in

Fig. 6e, f. As expected, in MERRA-2, the SAO winds are westerly on average during both QBO phases, while the SAO winds are easterly in the model control ensemble. SAO forcing terms at altitudes above the 1 hPa level show various discrepancies between MERRA-2 and the model (further discussed in Jaison et al., 2024 (in production)). However, it is readily noticeable that at 1hPa, the model lacks westerly GWD strength during both QBO phases.

The influence of the QBO phase on the SAO forcing terms is seen in Fig. 6a and b. Comparing Fig. 6 a and b, among the four forcing terms, the model meridional advection term appears to be least impacted by the QBO phase. Vertical advection, on the other hand, is substantially modulated by the QBO. In addition to the BDC, local induced circulations contribute to the vertical advection term. The BDC is known to consist of upwelling at equatorial latitudes throughout the year, while locally induced circulations will produce upwelling during periods of westward wave forcing (negative vertical wind shear)

and downwelling during periods of eastward wave forcing (positive vertical wind shear), thus maintaining approximate thermal wind balance. During the QBOE phase, winds at around 2hPa have a negative wind shear on average, thus creating an overall stronger upwelling in addition to the BDC upwelling. This then advects westerly winds from below, resulting in the positive vertical advection forcing seen in Fig. 6a above 10hPa.





For resolved waves and GWD, a direct response to the QBO is visible. When QBO westerlies are present in most of the low-mid stratosphere, more eastward waves are filtered out and thus at altitudes above 5hPa westward drag dominates. However, above 0.6 hPa, wave forcings are eastward on average in both QBO phases. This is consistent with the larger filtering of westward waves in the QBO region suggested by previous studies.

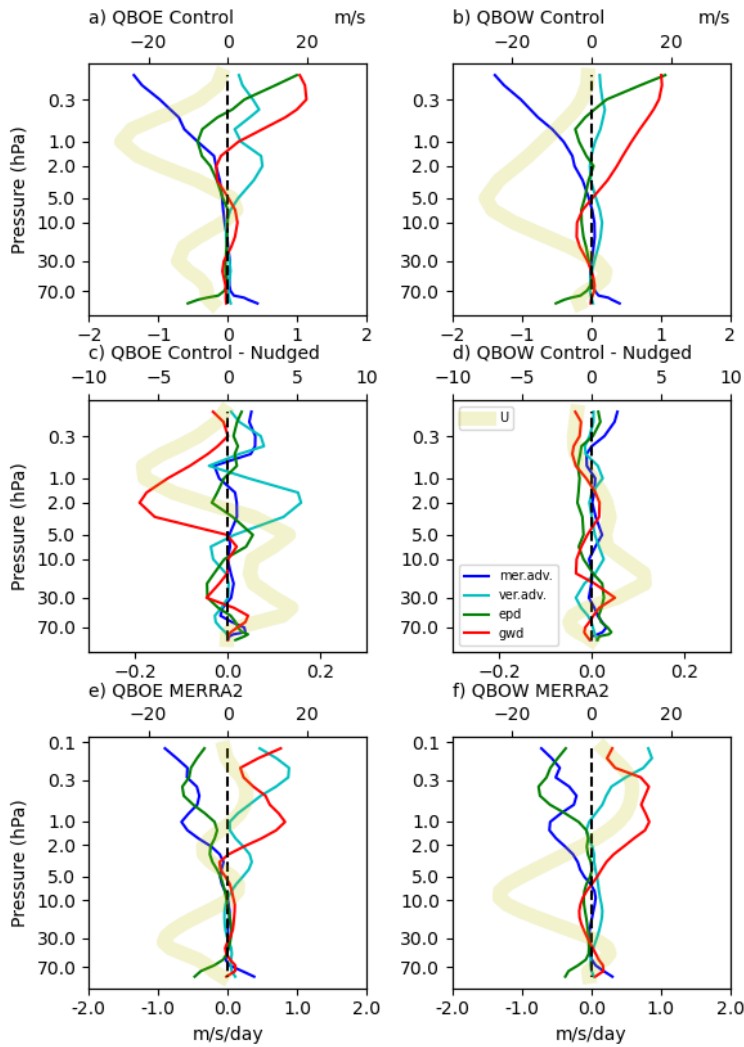


**Figure 6. Composite of time-mean TEM forcing terms (see eq 1, and legend in panel (d) ) and zonal-mean zonal winds ('U' thicker light-yellow line) averaged over 15S-15N for (a) Control experiment in QBOE months, (b) Control experiment in QBOW months, (c) Control minus Nudged differences during QBOE months, (d) Control minus Nudged differences during QBOW months, (e) MERRA-2 in QBOE months and (f) MERRA-2 in QBOW months. Units of TEM forcing terms are m/s/day (lower axis) and**

**zonal-mean zonal wind is m/s (upper axis).**





The major message from Fig. 6c,d is that changes in forcing as a result of the nudging are dominated by the GWD and vertical advection terms and these occur primarily in the QBOE phase (since these are approximately four times larger than during QBOW). Above 10 hPa during the QBOE phase, GWD and vertical advection show changes as large as 0.2 m/s/day on average. This is consistent with the results in Fig. 3a,b, where it was noted that the major bias in the QBO is during QBOE and thus the zonal winds are most altered during QBOE when nudging is applied.

Changes in zonal winds are directly reflected in the GWD profile. Nudging in the QBO height region led to increased QBO easterly winds, thus increasing the filtering of westward waves, resulting in the negative GWD difference values seen in Fig. 6c. Vertical advection changes are also a direct consequence of the zonal wind profile changes. As the westerlies at 10hPa weaken in the Nudged experiment during QBOE the vertical wind shear dampens, thus weakening the induced upwelling and creating the positive values of vertical advection changes in Fig. 6c.

Averaging of all months to form the QBO composite is slightly misleading for interpreting the EPD diagnostics, as a QBO phase typically lasts around a year or more, so the seasonal variations are averaged out in the composites. Figures 6c,d show that the differences in EPD forcing between the Control and Nudged experiments are small and independent of the QBO phase. However, Fig. 5 g-i showed that improvements associated with the EPD term are maximum during MAM. Seasonal QBO composites (not shown) have revealed that the main improvements associated with the EPD term occur in MAM primarily during the QBOE phase. The meridional advection term also suffers from the same averaging of seasonal variations, but Fig. 5 shows that the meridional advection contribution is small and insignificant. In summary, Fig. 6 highlights the individual QBO phase modulation of the SAO forcing terms and demonstrates that the nudging to reduce the QBO bias has most impact on the SAO forcing terms during the QBOE phase.

**4 Summary**

Modelling the upper stratosphere presents various challenges due to the limited availability of observations and the dependence on parameterization of small-scale processes. An easterly bias of the SAO has been reported in various climate models, suggesting that increased eastward wave forcing is required in the models. However, it is not clear if this underestimation of eastward wave forcing in the height region of the SAO is due to an underestimation of wave generation in the troposphere (e.g., associated with convection or frontogenesis) or whether there is excessive wave damping/filtering as the waves propagate vertically through the lower stratosphere, in the region dominated by the QBO (e.g., due to lower-level circulation biases). This study has investigated the latter and, specifically, whether reducing biases in the QBO winds can





lead to an improved representation of the SAO. This has been achieved using simulations of the HadGEM3 GA7.1 N216 that were performed as part of phase 2 of the QBOi model intercomparison project. SAO diagnostics have been compared from a
3-member ensemble of the free-running model (the Control experiment) and a corresponding 3-member ensemble in which the lower stratospheric winds in the height region of the QBO were nudged towards reanalysis to correct the well-known westerly zonal wind bias in the modelled QBO (the Nudged experiment).

The easterly bias of the SAO was found to be reduced in the Nudged experiment. The 42-year time-mean of equatorial zonal
mean zonal winds in Nudged changed as much as 25% compared to Control between 2 hPa and 0.6 hPa. A decrease in wind bias between 1 - 0.6hPa throughout the year indicates an improvement in both the SAO phases, i.e., a decrease of wind strength in the SAO easterlies and an increase in the SAO westerlies. The most significant reduction in the easterly SAO bias was during the transition from SAO westerlies to easterlies, with the westerlies persisting for longer in the Nudged experiment.


It was found that the QBO and SAO improvements are strongly coupled in the vertical. QBO composites (defined at 50hPa) showed that nudging towards the reanalysis produced the greatest QBO corrections when the 50 hPa QBO winds are in their easterly phase. This roughly coincides with the months when the 10 hPa QBO winds are in a westerly phase. An overall strengthening of the 50 hPa QBOE winds as well as the correction of the significant westerly bias in the 10 hPa westerly
winds are the most likely factors leading to this maximum correction of QBO winds during the QBOE phase. The diagnostics confirmed that this is the precise period during which the SAO bias was also improved.

The study further explored how the QBO correction impacted the processes that drive the SAO. QBO modulation of wave filtering, primarily during the QBOE phase, was found to be responsible for the major part of the SAO enhancement. Both
resolved waves and parameterized gravity waves contributions were enhanced during the equinoxes leading to enhanced SAO westerly phases. However, gravity wave forcing was found to play the major overall role, with the reduced westward forcing and improved eastward forcing in the nudged ensemble during periods of SAO westerly-to-easterly phase transitions, leading to longer and deeper SAO westerly phases and shorter easterly phases.

Changes in the advection forcing terms were also found. Except in the height range between 1 hPa to 0.6 hPa, vertical advection was found to counteract the SAO improvement by providing more westward forcing in the Nudged experiment. Meridional advection above the 0.6hPa level was found to strengthen as well, thus also counteracting the SAO improvements, especially in JJA. At all other levels changes in meridional advection forcing were small and insignificant. Since the BDC is the main contribution to the meridional advection, the QBO modification of advection is likely to originate
from an extratropical pathway, where changes in wave forcing cause corresponding changes in BDC. Exploring the details

of these extratropical routes is outside the scope of this study and the impacts are small compared to the tropical wave forcing changes.

The analysis presented in this study suggests that correcting biases in the lower altitude circulation alone is insufficient to completely mitigate all biases in the SAO. While correcting the underlying QBO wind bias has led to an improvement in wave filtering and thus the resulting representation of the SAO, there nevertheless remains a substantial easterly bias in the SAO. This suggests that enhanced momentum flux from high-frequency waves that are not absorbed in the QBO region are likely to be required to achieve a more accurate representation of the SAO. Such improvements might come through a better representation of tropospheric wave sources such as those associated with precipitation, convection and frontogenesis.


*Data Availability*

The model outputs used in this study is hosted by UK Centre for Environmental Data Analysis (CEDA) via JASMIN platform. MERRA-2 datasets used in this study are available at Global Modeling and Assimilation Office (GMAO) (2015), MERRA-2 tavg3_3d_asm_Nv: 3d,3-Hourly,Time-Averaged,Model-Level,Assimilation,Assimilated Meteorological Fields
V5.12.4, Greenbelt, MD, USA, Goddard Earth Sciences Data and Information Services Center (GES DISC), Accessed: [November 2022], 10.5067/SUOQESM06LPK.

*Acknowledgements*

AMJ thanks Oxford-Richards Graduate Scholarship for supporting this research. LJG and SO acknowledge funding from the
UK Research & Innovation (UKRI) and Natural Environment Research Council (NERC) through its funding of the National Centre for Atmospheric Science (NCAS) and the CANARI programme. JK was supported by the Met Office Hadley Centre Climate Programme funded by DSIT. MB was funded by the Met Office Climate Science for Service Partnership (CSSP) China project under the International Science Partnerships Fund (ISPF). Authors gratefully acknowledge the QBOi coordinators Neal Butchart, James Anstey, Yoshio Kawatani and Clara Orbe for their direction on the experiments and
diagnostics. Authors acknowledge with thanks UK Centre for Environmental Data Analysis (CEDA) and World Climate Research Programme (WCRP) Atmospheric Processes And their Role in Climate (APARC) for supporting QBOi project.

*Author contributions*

AMJ, LJG and SO designed the study, AMJ performed the analysis and produced all figures. LJG and SO provided
supervision. JK implemented the model nudging. MA and AMJ ran model experiments. AMJ wrote the original text with contributions from other authors.

*Competing interests.*

The authors declare that they have no conflict of interest.



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
