# Peer review of "Role of the Quasi-Biennial Oscillation on Alleviating Biases in the Semi-Annual Oscillation"

_EGUsphere, 2024_

## Author Comment (AC1)

Response to Reviewers

We thank both reviewers for their very valuable comments. We have made the suggested edits to the text and modified the figures accordingly. All changes are highlighted in the revised manuscript (using tracked changes - all revisions are shown in-line and deleted lines have strikethrough). Responses to the individual comments are provided below, with references to the corresponding line numbers in the highlighted manuscript.

Reviewer 1

In their paper "Role of the Quasi-Biennial Oscillation on Alleviating Biases in the Semi-Annual Oscillation" the authors investigate whether biases of the semiannual oscillation (SAO) in model simulations can be reduced by a more realistic quasi-biennial oscillation (QBO) in the stratosphere. For this purpose, free-running simulations of the HadGEM3 GA7.1 N216 atmosphere-only model ("Control") are compared with model simulations that are nudged to ERA-5 zonal-mean zonal winds in the QBO height region. Indeed, the SAO easterly bias can be reduced because the mode realistic QBO improves the filtering of upward propagating gravity waves. Still, some bias remains, possibly hinting at still too weak westerly wave forcing in the upper stratosphere and lower mesosphere.
Overall, the paper is very well written, the presented evidence supports the conclusions drawn, and the topic is of great relevance for the readership of WCD.

The paper is therefore recommended for publication in WCD after addressing my very minor comments as detailed below.

SPECIFIC COMMENTS
Fig.2: The distribution of phase transition months is quite different between MERRA-2 and the simulations. MERRA-2 has phase transitions in June and July, which is rarely seen in the nudged simulations, but has only one phase transition in February and March, which is much less probable than for the nudged and control simulations. Do you think this could have a significant effect on the characteristics of the composites?

Thank you for the comment. The difference in distribution of phase transition months in MERRA-2 compared to nudged simulations arises from the difference in altitude chosen to find the transition months. We have chosen 10 hPa for MERRA-2, while 5 hPa for model simulations.

However, we do not think this will have a significant effect on the characteristics of composites. In the composite plot (figure 3), below 10hPa, MERRA2 and nudged (which is nudged towards ERA5 data from 70 to 10hPa) behave quite similarly, confirming that the QBO characteristics are the same. In addition, our composites start either in March or September, preserving the semi-annual variabilities.

To eliminate any ambiguity related to the transition months in the first half of the year, a composite plot was created using only July-Dec data. The characteristics identified in both the current plot and the September-only composite plot are consistent.

l.354: Please be more specific! In Fig.5g and 5h eastward forcing by EPD is strongest at 0.2hPa around the solstices. However, the seasonality is different between 5 and 2hPa where weaker maxima of eastward forcing are found around the equinoxes.

Thank you so much for pointing this out. We have clarified the statement now – please see line 359.

TECHNICAL COMMENTS

Thank you so much for all the technical comments. We have resolved all of them now and we have noted the corresponding line numbers below.

l.64 and other occurrences:

reference Ern et al. (2015) is cited in the text, but is missing in the References
Ern et al. (2015) is now added in the References

l.79: reference Pena-Ortiz et al. (2008) is cited in the text, but is missing in the References

Pena-Ortez et al. (2008) is now added in the References

l.273: which figure? Fig.3?

Thank you, we have added the figure number in (see line 276).

l.333 resolve abbreviation HTE

Resolved in line 332.

l.353: please resolve abbreviation GWD

Added in line 192

l.354: please resolve abbreviation EPD

Added in line 191

l.370: in Nudged -> in the Nudged runs

Changed in line 368

l.370: compared to the Control -> compared to the Control runs

Changed in line 368

l.387: (Fig.6g-i) -> (Fig.5g-i)

Corrected in line 385

l.485: Nudged -> "Nudged"

Changed in line 497

l.485: Control -> "Control"

Changed in line 497

l.522: is hosted -> are hosted

Corrected in line 533

l.606-609: Please check reference Ern et al. (2023) for updates

Added reference for final version

l.619: Please check author list of Garfinkel et al. (2022)

Corrected

Reviewer 2

The article "Role of the Quasi-Biennial Oscillation on Alleviating Biases in the Semi-Annual Oscillation" investigates how biases in the model representation of the Quasi-Biennial Oscillation (QBO) impact the Semi-Annual Oscillation (SAO) in the stratosphere. The study highlights a common issue in models where the SAO exhibits a weaker westerly phase and stronger easterly phase compared to observations. It is noted that both resolved and parameterized tropical waves are too weak, affecting their propagation through the QBO-dominated region before reaching SAO altitudes. The research finds that correcting QBO biases can reduce the SAO easterly bias by improving the representation of these waves, with changes in zonal-mean winds at SAO altitudes reaching up to 25%. This correction also enhances the annual cycle in the equatorial upper stratosphere. However, despite these improvements, a significant easterly bias remains, indicating that westerly wave forcing in the upper stratosphere and lower mesosphere is still inadequately represented.
In my opinion, this manuscript is a highly valuable and intriguing piece of research. The results are well-interpreted and well-presented, making the article both engaging and informative. The clarity and depth of the analysis contribute significantly to the advancement of our understanding of the interactions between the QBO and SAO.

I strongly recommend this manuscript for publication in WCD. I have a few very minor technical comments that could enhance the readability and comprehension of the text for future readers.

**Specific Comments:**
It would be worthwhile to improve the layout of the panels and the font sizes in the Figures of the article. Figures 3-5 lack units on the color scales. I would also suggest a better arrangement of the panels in the Figures (spacing them out). For example, in Figure 4, the labels "Latitude" on the x-axes in panels a) and b) almost look like the titles of panels c) and d). Figure 6 is the hardest to read due to the panel layout and small font sizes in the legends. I would recommend enlarging this figure so that the axis labels and panel titles are not so close to each other. Additionally, it would be helpful to increase the font size—especially in the legends—to make them easier to read.

Thank you so much for the suggestions to improve the figures. We have improved the panel spaces, legend sizes and added units in figures 3-5. The revised draft has new figures added.

**Technical Comments:**
- Line 56: NH has been already defined in line 31.

  Corrected

- Line 70: What is GW?

  Defined GWD in line 191

- Line 104: QBOE has already been defined in line 67

  Corrected

- Line 120: "...from 2015-2020...." – Using "to" instead of a hyphen may be better.

  Changed in line 121

- Line 159: "...(Ern et al., 2023) the SAO..." – missing coma before "the"

  Added in line 155

- Lines 181-192: I would suggest adding explanations for the remaining quantities (a, f, $\rho_0$, $\phi$).
  Added the suggested explanations in lines 179 to 181.

- Line 259: QBOW has not been yet defined.

  Added in line 268

- Line 273: "…that the Fig. shows…" – Please add the number of the Figure.

  Corrected in line 276

- Line 327: "…denotes at 95%…" – "at" should be removed.

  Corrected

- Line 333: What is THE?

  Changed to HT effect in line 332

- Line 343: "H-T" – previously Holton-Tan relationship was indicated as HT (line 333). Please unify the indications.

  Corrected

- Line 353: What is GWD?

  Defined GWD in line 192

- Line 354: What is EPD?

  Defined EPD in line 191

- Line 387: There are no "g-i" panels on Figure 6.

  Changed to Fig 5 in line 385

- Line 522: "…study is hosted…" – should be "are"

  Corrected in line 533